# Caspase-1 Inhibition Reduces Occurrence of PANoptosis in Macrophages Infected by *E. faecalis* OG1RF

**DOI:** 10.3390/jcm11206204

**Published:** 2022-10-21

**Authors:** Danlu Chi, Yuejiao Zhang, Xinwei Lin, Qimei Gong, Zhongchun Tong

**Affiliations:** 1Hosiptal of Stomatology, Sun Yat-sen University, Guangzhou 510055, China; 2Guangdong Provincial Key Laboratory of Stomatology, Sun Yat-sen University, Guangzhou 510055, China; 3Guanghua School of Stomatology, Sun Yat-sen University, Guangzhou 510055, China

**Keywords:** caspase-1, *Enterococcus faecalis*, macrophage, PANoptosis

## Abstract

To investigate the effect of caspase-1 inhibition on PANoptosis in macrophages infected with *Enterococcus faecalis* OG1RF. RAW264.7 cells with and without pretreatment by caspase-1 inhibitor were infected with *E. faecalis* OG1RF at multiplicities of infection (MOIs). A live cell imaging analysis system and Western blot were applied to evaluate the dynamic curve of cell death and the expression of executor proteins of PANoptosis. The mRNA expression of *IL-1**β* and *IL-18* was quantified by RT-qPCR. Morphological changes were observed under scanning electron microscopy. We found that PI-positive cells emerged earlier and peaked at a faster rate in *E. faecalis*-infected macrophages (Ef-MPs) at higher MOIs. The expression of the N-terminal domain of the effector protein gasdermin D (GSDMD-N), cleaved caspase-3 and pMLKL were significantly upregulated at MOIs of 10:1 at 6 h and at MOI of 1:1 at 12 h postinfection. In Ef-MPs pretreated with caspase-1 inhibitor, the number of PI-positive cells was significantly reduced, and the expression of *IL-1β* and *IL-18* genes and cleaved caspase-1/-3 and GSDMD-N proteins was significantly downregulated (*p* < 0.05), while pMLKL was still markedly increased (*p* < 0.05). Ef-MPs remained relatively intact with caspase-1 inhibitor. In conclusion, *E. faecalis* induced cell death in macrophages in an MOI-dependent manner. Caspase-1 inhibitor simultaneously inhibited pyroptosis and apoptosis in Ef-MPs, but necroptosis still occurred.

## 1. Introduction

*Enterococcus faecalis* is a member of the human commensal microbiota but is one of the common causes of hospital-acquired infections [1]. *E. faecalis* utilizes a series of virulence factors, such as lipoteichoic acid (major cell wall constituent), esp (protein surface), ace (collagen binding protein), gelE (gelatinase) and cylA (hemolysin activator), for adhesion and invasion, modulating the immune response of the host [2,3,4,5]. *E. faecalis* has a strong survival ability and resists starvation, antibiotics, bile salt and various adverse environments [6,7,8,9]. In dentistry, *E. faecalis* is also a microorganism commonly detected in persistent intraradicular infections after failure of endodontic treatments [10,11], and immune responses of periapical tissue caused by *E. faecalis* have become an interesting topic over the last few years.

The immune system has developed multiple mechanisms to restrict microbial infections and regulate inflammatory responses. Recently, multiple lines of evidence have indicated that a unified mechanism named PANoptosis is formed among apoptosis, pyroptosis and necroptosis through regulatory proteins and transmission pathways [12,13,14]. PANoptosis is involved in infectious and autoinflammatory diseases, cancer and other conditions and is regulated by the PANoptosome, which provides a molecular scaffold that allows for interactions and activation of the machinery required for inflammasome/pyroptosis, apoptosis and necroptosis [15,16,17,18,19,20]. Caspase-1 is a component of the PANoptosome and is considered to play a primary role in PANoptosis [12,13]. Caspase-1 cleaves the pore-forming protein gasdermin D (GSDMD), causes oligomerization of the N-terminal portion of GSDMD, forms a pyroptotic pore and finally results in pyroptosis [21,22]. Macrophages are one of the frontline defense cells of the human immune system and are critical to both acute and resolving immune responses by initiating programmed cell death pathways. The evolution of bacteria helps to evade immune attack, but macrophages managed to recognize and clear the evolutional pathogens. A few studies indicated that PANoptosis may be activated when macrophages are stimulated by a wide range of pathogens, including bacteria, viruses, fungi and parasites [12,15,16,23,24].

It is important to clarify the pathogenic mechanism of *E. faecalis*-infected macrophages due to their involvement in urinary tract infections, hepatobiliary sepsis, endocarditis, surgical wound infections, bacteraemia, neonatal sepsis and the prevention of other hospital-acquired infections [25,26]. A previous study showed the *E. faecalis* induced different levels of expression of apoptosis, pyroptosis and necroptosis that were probably associated with PANoptosis in macrophages [27]. In this study, we further investigated the real-time effect of *E. faecalis* on pyroptosis, apoptosis and necroptosis of macrophages at different MOIs and evaluated the effect of caspase-1 inhibitor on PANoptosis in macrophages infected by *E. faecalis*.

## 2. Materials and Methods

### 2.1. Culture of Enterococcus faecalis and Macrophages

*E. faecalis* OG1RF and RAW264.7 murine macrophage cell lines (ATCC, Manassas, VA) were used in this study. *E. faecalis* OG1RF was routinely streaked and grown on a brain heart infusion agar (BHI; Difco, Detroit, MI, USA) under aerobic conditions at 37 °C. A single colony was inoculated into 5 mL of BHI broth and grown overnight to reach the exponential phase, which corresponds to a bacterial concentration of approximately 10^9^ cfu/mL. The RAW264.7 murine macrophage cell line was cultured in alpha-minimal essential medium (α-MEM; Gibco, New York, NY, USA) with 10% fetal bovine serum (FBS; Gibco, New York, NY, USA) in a 5% CO_2_ humidified incubator at 37 °C. RAW264.7 cells were seeded overnight in 96-well culture plates at a density of 7 × 10^3^/well for real-time cell death analysis, in 12-well culture plates at 3 × 10^4^/well for scanning electron microscopy and in 10 cm dishes at 3 × 10^6^/well for quantitative real-time polymerase chain reaction (RT-qPCR) and Western blotting.

### 2.2. RAW264.7 Cells Infected with E. faecalis

RAW264.7 cells were infected with *E. faecalis* OG1RF at different multiplicities of infection (MOIs) and cultured in a humidified incubator with 5% CO_2_ at 37 °C for subsequent experiments. To study the inhibition of caspase-1, RAW264.7 cells were first pretreated with caspase-1 inhibitor Ac-YVAD-CMK (YVAD, 30 μmol/L, GLPBIO, Montclair, CA, USA) for 30 min and were then infected with *E. faecalis* OG1RF at an MOI of 100:1 to evaluate the effect of caspase-1 inhibitor on *E. faecalis*-infected macrophages. In the evaluation of caspase-1 inhibitor, RAW264.7 cells alone and with only pretreatment by YVAD were referred to as the control group and YVAD-control group, respectively. *E. faecalis*-infected RAW264.7 cells without and with the inhibitor YVAD were named Ef-MP and YVAD-Ef-MP, respectively.

### 2.3. Real-Time Cell Death Analysis

Real-time cell death analysis was used to depict the dynamic and morphological changes of macrophages with *E. faecalis* infection. After staining with 0.5 μg/mL propidium iodide dye (PI, Solarbio, Beijing, China) for 30 min, RAW264.7 cells were infected with *E. faecalis* OG1RF at MOIs of 1000:1, 500:1, 200:1, 100:1, 50:1, 10:1 and 1:1, and real-time cell death was evaluated from 0 to 22 h. RAW264.7 cells with only the addition of 1% Triton were used as a positive control for PI staining. Furthermore, to examine the effect of caspase-1 inhibitor on the death of macrophages infected with *E. faecalis*, RAW264.7 cells were infected with *E. faecalis* at an MOI of 100:1 after pretreatment with Ac-YVAD-CMK. The number of PI-positive cells in each well was quantified per hour using a live cell imaging analysis system (BioTek, Winooski, VT, USA), and a real-time cell death dynamic curve was depicted.

### 2.4. Quantitative Real-time Polymerase Chain Reaction (RT-qPCR)

Total RNA was extracted and purified from RAW264.7 cells at 6 h and 12 h postinfection using an RNA-Quick purification kit (YISHAN Biotechnology, Shanghai, China). After spectrophotometric quantification by a Nanodrop 2000 spectrophotometer (Thermo Scientific, Waltham, MA, USA), total RNA was synthesized into cDNA using a reverse transcription kit (Takara, Kyoto, Japan). RT-qPCR was performed with a QuantStudio 5 real-time PCR machine (Applied Biosystems, Thermo Fisher, Waltham, MA, USA) with a SYBR qPCR Supermix Plus kit (Novoprotein, Shanghai, China). Relative expression of *IL-1β* and *IL 18* was normalized to that of ß-actin according to the 2^−ΔΔCt^ method. The following primer sequences were used: *IL-1β*: Forward, 5′-CCAAAAGATGAAGGGCTGCTT-3′, Reverse, 5′- GAAAAGAAGGTGCTCATGTCCTC-3′; *IL-18*: Forward, 5′- TGTCAGAAGACTCTTGCGTCAAC-3′, Reverse, 5′- GATTCCAGGTCTCCATTTTCTTCA-3′; *β-actin*: Forward, 5′-GGCTGTATTCCCCTCCATCG-3′, Reverse, 5′-CCAGTTGGTAACAATGCCATGT-3′.

### 2.5. Western Blotting

Total protein from RAW 264.7 cells was harvested at 6 h and 12 h postinfection and quantified using a BCA protein assay kit (Beyotime, Shanghai, China) following the manufacturer’s instructions. Proteins were separated on a 12% SurePAGE precast gel and electrotransferred to PVDF membranes. After being blocked in 5% nonfat milk at room temperature for 1 h, membranes were incubated with primary antibodies overnight at 4 °C, including ß-actin (1:1000, Cell Signaling Technology [CST], Danvers, MA, USA, #4970), caspase-3 (1:1000, CST, #9662), caspase-1 (1:1000, CST, #89332), GSDMD (1:1000, Abcam, ab209845), pMLKL (1:1000, CST, #37333) or MLKL (1:1000, CST, #37705). Subsequently, membranes were incubated with horseradish peroxidase (HRP)-conjugated goat anti-rabbit secondary antibody (1:5000) for 1 h at room temperature. Protein bands were visualized using an ECL kit (Millipore, Bedford, MA, USA) under the Image Quant LAS 4000 mini imaging system and protein levels were quantified using ImageJ software.

### 2.6. Scanning Electron Microscopy

Scanning electron microscopy was used to observe the morphological changes of macrophages infected by *E. faecalis* OG1RF at an MOI of 100:1, with and without the caspase-1 inhibitor Ac-YVAD-CMK. At 6 h and 12 h postinfection, RAW 264.7 cells were harvested and fixed with 2.5% glutaraldehyde overnight at 4 °C. After being rinsed with PBS three times, the specimens were dehydrated through a graded series of ethanol (30, 50, 70, 80, 95 and 100%) and dried using tertiary butanol. The dried specimens were sputter coated with gold-palladium after freezing for more than 3 h and then observed with a scanning electron microscope (Quanta 400F, FEI, Brno, Czech Republic) operating at 10 kV.

### 2.7. Statistical Analysis

All experiments were repeated at least three times and data are presented as the mean ± standard deviation (SD). One-way ANOVA was used to analyze data that conformed to the normal distribution and homogeneity of variance between multiple groups, followed by Tukey’s post hoc test using SPSS 25.0. Uneven variance data were analyzed with the Kruskal–Wallis test. The level of significance was set as *p* < 0.05.

## 3. Results

### 3.1. Dynamic Changes in Macrophages Infected with E. faecalis in a Dose-Dependent Manner

In the real-time cell death analysis, the dynamic death curve of *E. faecalis*-infected RAW264.7 cells correlated with an increasing bacterial MOI in a dose-dependent manner. PI-positive cells represented the emergence of cell death. The higher the initial MOI was, the earlier PI-positive cells emerged and the more rapidly the peak PI-positive cell value was reached. The peak of PI-positive cells occurred later at an MOI of less than 200:1, but was higher than that of the positive control group. In contrast, the peak of PI-positive cells was lower than that of the positive control group beyond an MOI of 200:1, despite the earlier occurrence (Figure 1a). In dynamic cell death, morphological changes in apoptosis, pyroptosis and necroptosis were observed (Figure 1b–g). Apoptotic cells showed shrinking cell bodies and classic apoptotic bodies. Necroptosis presented significant swelling of the cell body, followed by an explosion of the cell body such as an overinflated balloon accompanied by PI uptake. Pyroptotic cells displayed little swelling and formed multiple bubble-like protrusions in conjunction with PI staining before rupture of the plasma membrane.

### 3.2. Protein Expression of Macrophages Infected by E. faecalis at Three MOIs

Executor proteins of apoptosis, pyroptosis and necroptosis were estimated in *E. faecalis*-infected macrophages at initial MOIs of 1:1, 10:1 and 100:1 (Figure 2). Cleaved caspase-3, GSDMD-N and pMLKL were significantly upregulated at 6 h and 12 h in *E. faecalis*-infected RAW264.7 cells at MOIs of 100:1 and 10:1 in comparison with the control group (*p* < 0.05). At an MOI of 1:1, the expression of the three executor proteins did not show a significant difference at 6 h postinfection compared to the control group (*p* > 0.05), but the executor proteins showed higher expression than the control group at 12 h postinfection (*p* < 0.05).

### 3.3. The Effect of Caspase-1 Inhibitor on Real-Time Cell Death of Macrophages

In the dynamic death curve at an MOI of 100:1, the emergence of PI-positive cells was delayed by the caspase-1 inhibitor in *E. faecalis*-infected RAW 264.7 cells and the number of PI-positive cells was significantly lower with the inhibitor YVAD than without the inhibitor YVAD at the same time point of infection. PI-positive cells began to rise in *E. faecalis*-infected cells at 6 h and under the inhibitor YVAD, PI-positive cells ascended until 12 h (Figure 3a). Large amounts of red-stained PI-positive cells were detected among *E. faecalis*-infected cells, while little red staining was perceived in the presence of inhibitor YVAD (Figure 3b).

### 3.4. The Effect of Caspase-1 Inhibitor on Gene Expression of Two Cytokines

Six and twelve hours of *E. faecalis* infection significantly improved the mRNA expression of inflammatory factors *IL-1β* and *IL-18* in RAW 264.7 cells compared with the control group (*p* < 0.05), but the high expression of *IL-1β* and *IL-18* was significantly inhibited in the presence of the caspase-1 inhibitor (Figure 4).

### 3.5. Protein Expression of E. faecalis-Infected Macrophages Treated with a Caspase-1 Inhibitor

Executor proteins of apoptosis, pyroptosis and necroptosis were examined in *E. faecalis*-infected macrophages with and without the caspase-1 inhibitor (Figure 5). The protein expression of cleaved caspase-1, GSDMD-N, cleaved caspase-3 and pMLKL was increased at 6 and 12 h in *E. faecalis*-infected RAW264.7 cells at an MOI of 100:1 (Figure 5b–e). The upregulation of cleaved caspase-1, GSDMD-N and cleaved caspase-3 was significantly inhibited by the inhibitor YVAD (Figure 5b–d) and the expression of pMLKL was not obviously reduced under the inhibitor YVAD in *E. faecalis*-infected RAW264.7 cells (Figure 5e). Compared with the control group, the expression of cleaved caspase-1, GSDMD-N and cleaved caspase-3 showed no significant difference in the YVAD-Ef-MP group, but the expression of pMLKL was increased (*p* < 0.05).

### 3.6. Morphological Modification of E. faecalis-Infected Macrophages under Caspase-1 Inhibitor Treatment

In the morphological observation of *E. faecalis*-infected macrophages, the three classic forms of PANoptosis, apoptosis, pyroptosis and necroptosis, were visualized. Apoptotic cells exhibited wrinkled cell bodies and multiple apoptotic bodies (Figure 6b). Late pyroptotic cells showed cytomembrane rupture and a slightly swollen cell body (Figure 6c). Necroptotic cells showed obviously swollen cell bodies and finally resulted in an explosion of the cell body. Some *E. faecalis* were observed on the remaining cell components at 12 h postinfection (Figure 6d). Morphological changes in the inhibitor’s protective effect on RAW264.7 cells infected with *E. faecalis* at 6 and 12 h postinfection were also observed under SEM (Figure 6e–h). *E. faecalis*-infected RAW 264.7 cells showed multiple morphological lesions (Figure 6e,f), whereas *E. faecalis*-infected RAW 264.7 cells maintained a relatively complete cell morphology under YVAD inhibitor pretreatment (Figure 6g,h).

## 4. Discussion

*E. faecalis* infections have become difficult to treat due to their intrinsic and acquired resistance to a range of antibiotics and their ability to evade immune cell clearance and host immune cells have also evolved complex feedback mechanisms against *E. faecalis* [26,28]. Multiple mechanisms are interconnected to sense abnormalities in key proteins in their signaling pathways and initiate the assembly of a variety of cell death complexes against infections [29,30,31,32,33]. The currently available clinical evidence suggests that root canal chemomechanical preparation techniques do not thoroughly eliminate bacteria during root canal treatment [34] and the few residual bacteria in the root canal may not always cause periapical infection. Furthermore, we found that periapical inflammation did not necessarily occur for teeth even with inadequate root canal treatment. This phenomenon indicates that the quantity of pathogenic bacteria and feedback of the host decide inflammation occurrence in periapical tissue. *E. faecalis* is a microorganism commonly detected in retreated root canals [10] and the amount of *E. faecalis* may also affect the immune response of periapical tissue. In a study by Zou et al. [35], when macrophages were induced by a wide spectrum of proapoptotic stimuli, *E. faecalis* strain E99 (a clinically isolated strain from the urine of a patient) at an MOI of 10:1 might disturb the occurrence of apoptosis. In the study by Mohamed Elashiry et al. [2]., *E. faecalis* infection at a low MOI of 1:1 in bone marrow stem cells (BMSCs) for a prolonged time reduced apoptotic activity in subsequently differentiated macrophages. The two studies indicated that the low quantity of *E. faecalis* infection can escape the immune attack of macrophages, interfere with the apoptosis of macrophages and prolong their intracellular survival time. In our study, we also found that the emergence of cell death in the low MOI group was obviously later than that in the high MOI group. The expression of the apoptosis executor protein cleaved caspase-3 at an MOI of 1:1 at 6 h postinfection showed no significant difference compared to the control group, whereas cleaved caspase-3 was significantly upregulated at an MOI of 1:1 at 12 h postinfection and at MOIs of 10:1 and 100:1 at both 6 and 12 h postinfection. Moreover, the expression of the executor protein GSDMD-N in pyroptosis and the executor protein pMLKL in necroptosis were also consistent with the caspase-3 results. According to the results, a low quantity of *E. faecalis* did not cause PANoptosis in macrophages, which was one of the reasons that the residual bacteria in the root canal did not give rise to periapical diseases. However, when root canal orifices are open and nutrition in oral cavity freely enters the periapical tissue, the environment will be suitable for bacterial proliferation. The residual *E. faecalis* grew a certain threshold and might induce PANoptosis. Therefore, residual bacteria still have the potential to cause periapical diseases.

In the conflict between pathogenic bacteria and the host immune system, as one of the corresponding feedback mechanisms of host evolution, PANoptosis is critical for restricting a wide range of pathogens, such as bacteria, viruses, fungi and parasites [12,13]. During the process of PANoptosis against microbial infections, inflammatory cell death occurs via the joint activation of pyroptosis, apoptosis and necroptosis, which avoids pathogen-mediated inhibition of individual death pathways. PANoptosis provides a mechanism for the host to activate alternative cell death defense mechanisms if pyroptosis, apoptosis and necroptosis are compromised by a pathogen or other blockade [36]. A recent study has shown that caspase-1 is both a regulated protein of pyroptosis and a key catalytic effector in the PANoptosome and plays an important role in regulating and catalyzing downstream effector proteins in the occurrence of PANoptosis [36,37]. Ac-Val-Ala-Asp-CMK (Ac-YVAD-CMK) is a classic inhibitor of caspase-1 that can significantly reduce the protein levels of caspase-1 (p20, the cleaved-caspase-1) in a low dose and inhibits the maturation of *IL-1β* [38,39,40]. Ac-YVAD-CMK had no inhibitory effect on either the apoptosis induced by ginsenoside-Rh2 or the necrosis and apoptosis induced by shikonin, suggesting that Ac-YVAD-CMK alone could not inhibit apoptosis and necrosis [41,42]. However, Ac-YVAD-CMK inhibited both pyroptosis and apoptosis in macrophages induced by LPS, which was in contrast to the PANoptosis signaling pathway [43]. Therefore, Ac-YVAD-CMK was considered a caspase-1 inhibitor to study the effect of inhibition of caspase-1 on PANoptosis in macrophages infected by *E. faecalis* in this study.

In our study, a caspase-1 inhibitor showed a certain inhibitory effect on the death of macrophages infected by *E. faecalis*. The dynamic death curve showed that the number of PI-positive cells significantly decreased in *E. faecalis*-infected macrophages pretreated with the caspase-1 inhibitor and almost no dead cells emerged within 12 h as in the control group. Morphological observation indicated that *E. faecalis*-infected macrophages maintained relatively normal cell morphology under the caspase-1 inhibitor. Furthermore, the mRNA expression of two cytokines, *IL-1β* and *IL-18*, in macrophages was reduced under the caspase-1 inhibitor. Among the executor proteins of PANoptosis, the expression of cleaved caspase-1/-3 and GSDMD-N was significantly downregulated under the caspase-1 inhibitor and almost equalled that in the blank control group. However, the expression of pMLKL did not significantly decrease and was markedly higher than that in the blank control group. These findings indicated that the caspase-1 inhibitor could simultaneously inhibit the occurrence of pyroptosis and apoptosis under the collective mechanism of PANoptosis and reduce cell death, but it had little effect on the occurrence of necroptosis. Additionally, we found that the number of PI-positive cells treated with the caspase-1 inhibitor began to increase at 12 h postinfection, which might be related to the activation of necroptosis. The results indicated that the inhibition of caspase-1 hindered the pyroptosis and apoptosis pathways of PANoptosis and the partial inhibition did not prevent the overall augmentation of cell death.

Pyroptosis, necroptosis and apoptosis seem to be noncorrelated forms of programmed cell death, but they form multilevel interactions through the assembly of PANoptosomes [13,36,44]. The interference between molecules in any pathway will affect the execution of other pathways. The expression of pMLKL was clearly increased and necroptosis was activated when the pyroptotic proteins caspase-1 and GSDMD were deficient in response to fungal pathogens [15]. Kuriakose et al. found that influenza A virus (IAV) infection strongly activated caspase-1/-8 in BMDMs with *MLKL* deficiency, but the addition of a caspase-8 inhibitor prevented IAV-infected cell death. Treatment with a caspase-8 inhibitor plus an MLKL inhibitor prevented wild-type BMDMs from undergoing cell death during IAV infection [18]. Similarly, in BMDMs infected by vesicular stomatitis virus (VSV), *Listeria monocytogenes* or *Salmonella enterica serovar Typhimurium* in the study of Christgen et al., knockdown of *caspase-1/-11* only reduced the cell death of *Salmonella*-infected BMDMs and did not decrease the cell death of BMDMs infected by *Listeria*, IAV or VSV. The combined knockdown of *caspase-1/-11/-8* and *RIPK3* could protect macrophages from cell death induced by the four pathogens to a large extent [16]. These studies showed that there were multiple concurrent pathways of cell death caused by microbial infections and there was a close relationship between these pyroptosis, necroptosis and apoptosis. Moreover, the different types of pathogens showed individual characteristics in the induction of PANoptosis. PANoptosis occurs on a pathogen-specific basis and may be predicated on the ability of individual pathogens that are highly virulent to evade frontline innate immune defense [13]. *E. faecalis* can cause infectious diseases but is also part of the normal flora of the gastrointestinal tract. *E. faecalis* in the oral cavity does not possess high pathogenicity in comparison with the pathogenic fungi, viruses and bacteria mentioned above and thus the low quantity of *E. faecalis* might prolong the occurrence of PANoptosis in macrophages. In this study, we only used a caspase-1 inhibitor to study PANoptosis. However, the mechanisms of the interrelation and interaction among signal pathways of PANoptosis are very complex and more signal pathways need to be further studied. Moreover, caspase-3 and caspase-1 were considered ones of the components of PANoptosome in *E. faecalis*-infected macrophages, but their exact roles in the signal pathway need to be further verified. Future research would like to focus on the mechanism to clarify the interaction among the compositions of PANoptosome when *E. faecalis*-infected macrophages develop PANoptosis.

## 5. Conclusions

Macrophage infected by *E. faecalis* was related to MOI in a dose-dependent manner and *E. faecalis* infection at higher MOI resulted in quicker macrophage death. In *E. faecalis*-infected macrophages, a caspase-1 inhibitor could simultaneously inhibit the pyroptosis and apoptosis pathways of PANoptosis to reduce cell death, but necroptosis was still activated to cause subsequent cell death.

## Figures and Tables

**Figure 1 jcm-11-06204-f001:**
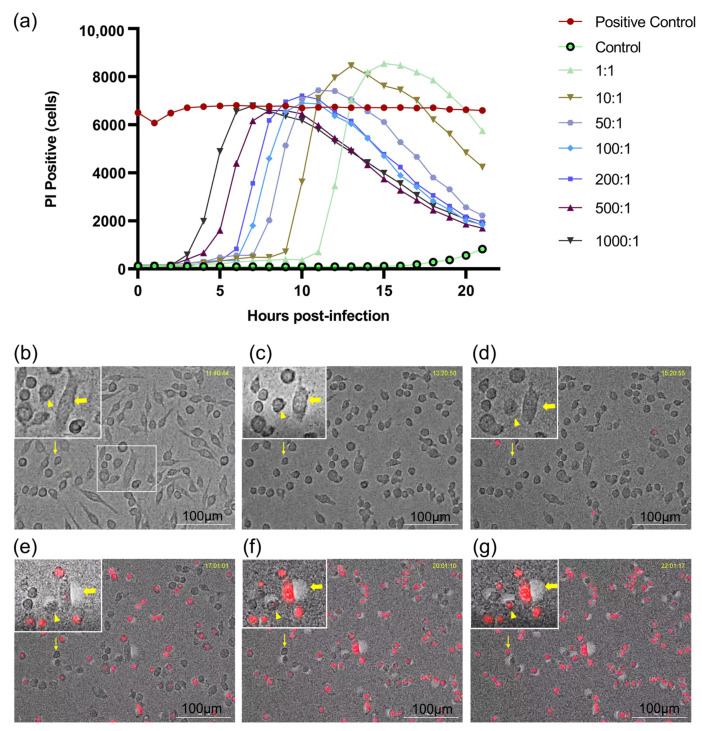
Real-time cell death analysis of macrophages infected by *Enterococcus faecalis*. (**a**) Dynamic cell death curve of RAW 264.7 cells infected with *E. faecalis* OG1RF at different MOIs (MOI = 1:1, 10:1, 50:1, 100:1, 200:1, 500:1 and 1000:1); (**b**–**g**) Representative time-lapse images of RAW 264.7 cells infected with *E. faecalis* OG1RF. Apoptosis (thin arrow), pyroptosis (triangle) and necroptosis (thick arrow) were inferred to occur with dynamic changes of cells. Cell morphology was visualized by microscopy (20×) and red fluorescence indicated PI staining. Scale bar: 100 μm.

**Figure 2 jcm-11-06204-f002:**
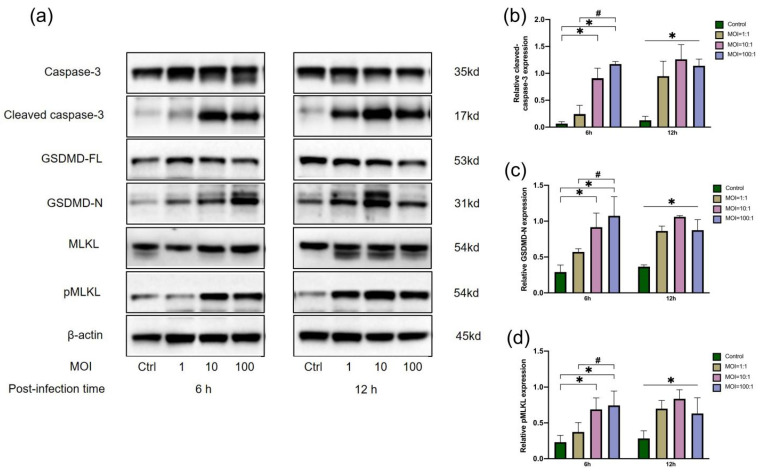
Relative expression of executor proteins of apoptosis, pyroptosis and necroptosis in RAW 264.7 cells infected by *E. faecalis* at different MOIs (as indicated). Representative immunoblotting bands (**a**) and quantification indicated an upregulation of cleaved caspase-3 (**b**), GSDMD-N (**c**) and pMLKL (**d**) using western blot. * *p* < 0.05 compared to the control group. # *p* < 0.05 compared to the other experimental groups.

**Figure 3 jcm-11-06204-f003:**
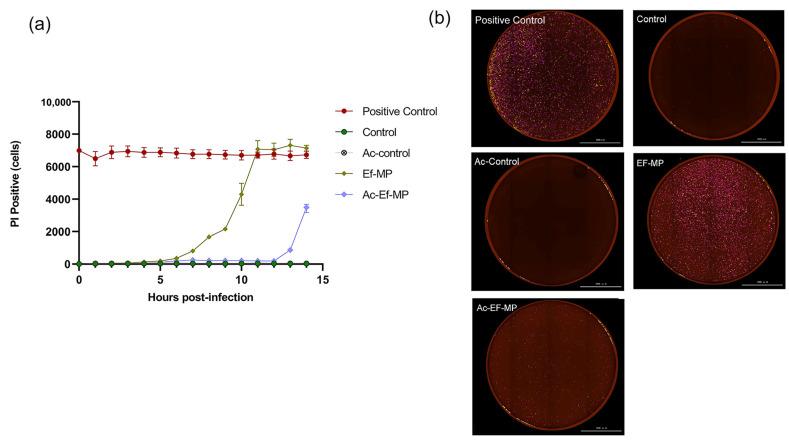
The impact of caspase-1 inhibitor on the cell death of macrophages infected by *E. faecalis*. (**a**) Quantification of PI-positive cells over time in *E. faecalis*-infected RAW 264.7 cells. (**b**) All samples were stained with PI and captured at the peak in a live cell imaging analysis system (BioTek, Winooski, VT, USA). Control: RAW 264.7 alone; YVAD-Control: RAW 264.7 only pretreated with inhibitor YVAD; Ef-MP: *E. faecalis*-infected RAW 264.7; YVAD-Ef-MP: RAW 264.7 pretreated with inhibitor YVAD and then infected by *E. faecalis*. All the figures in (**b**) were captured at 12 h postinfection. Scale bar: 2000 μm.

**Figure 4 jcm-11-06204-f004:**
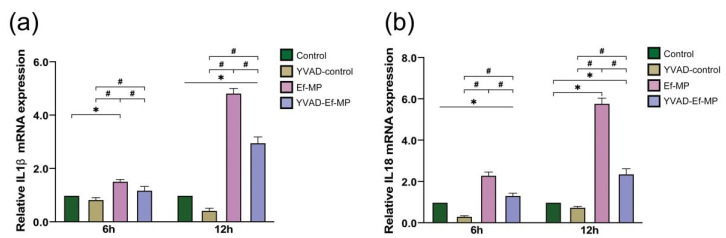
The mRNA expression of two cytokines under caspase-1 inhibitor treatment. The relative expression of *IL-1β* (**a**) and *IL-18* (**b**) with and without the inhibitor YVAD was examined in *E. faecalis*-infected RAW 264.7 cells at 6 h and 12 h postinfection by RT-qPCR. RAW264.7 cells alone and cells pretreated with the inhibitor YVAD were referred to as the control group and YVAD-control group, respectively. *E. faecalis*-infected RAW264.7 cells without and with the inhibitor YVAD were named Ef-MP and YVAD-Ef-MP, respectively. * *p* < 0.05 compared to the control group. # *p* < 0.05 compared to the Ef-MP group.

**Figure 5 jcm-11-06204-f005:**
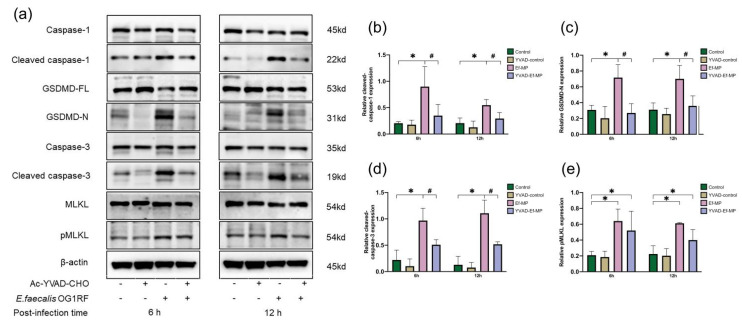
The expression of proteins related to PANoptosis under caspase-1 inhibitor treatment. Relative expression of the proteins related to PANoptosis with and without the inhibitor Ac-YVAD-CMK was examined in *E. faecalis*-infected RAW264.7 cells at 6 h and 12 h postinfection with Western blot. RAW264.7 cells alone or with pretreatment with the inhibitor YVAD were referred to as the control group and YVAD-control group, respectively. *E. faecalis*-infected RAW264.7 cells without and with the inhibitor YVAD were named Ef-MP and YVAD-Ef-MP, respectively. Representative immunoblotting bands of four proteins (**a**) and quantitative analysis of cleaved caspase-1 (**b**), GSDMD-N (**c**), cleaved caspase-3 (**d**) and pMLKL (**e**) by Western blot. * *p* < 0.05 compared to the control group. # *p* < 0.05 compared to the Ef-MP group.

**Figure 6 jcm-11-06204-f006:**
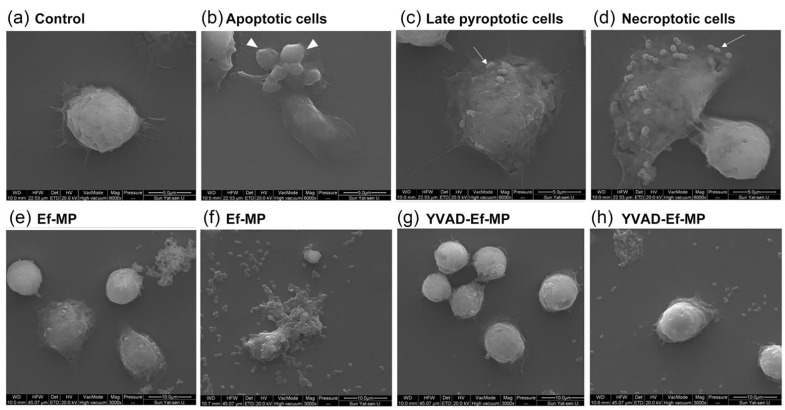
Representative scanning electronic microscopy (SEM) images of RAW 264.7 cells infected with *E. faecalis* OG1RF. (**a**) Control RAW264.7 cells; (**b**) Apoptotic cells: classic apoptotic bodies (white arrowhead); (**c**) Late pyroptotic cells and (**d**) Necroptotic cells. Some *E. faecalis* can be seen on the remaining cell bodies on both pyroptotic and necroptotic cells (white arrow). Scale bar: 5 μm. *E. faecalis*-infected RAW 264.7 cells showed different morphological damage (**e**,**f**); *E. faecalis*-infected RAW 264.7 cells had a relatively complete cell morphology under the inhibitor YVAD pretreatment and few dead cells could be seen (**g**,**h**). Scale bar: 10 μm.

## Data Availability

Not applicable.

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
