# Peer review of "Caspase-1 Inhibition Reduces Occurrence of PANoptosis in Macrophages Infected by E. faecalis OG1RF"

_jcm, 2022, doi:10.3390/jcm11206204_

Round 1

Reviewer 1 Report

This manuscript presents an interesting study for the effect of Caspase-1 inhibition on reduction of PANoptosis in E. faecalis infected macrophages. The results showed that there are significant differences in the number of dead cells and cell morphology between with and without Ac-YVAD-CMK for 6/12 hours incubation after infection. The data and the methods may be of use for dental and medical fields in terms of bacterial infection control. However, I have several concerns that need to be addressed.

#1. In page 2, Materials and Methods section, the sentence “. RAW264.7 cells were seeded overnight in 96-well culture plates at a density of 7×103/well for real-time cell death analysis, in 12-well culture plates at 3×104/well for scanning electron microscopy, and in 10-cm dishes at 3×106/well for quantitative real-time polymerase chain reaction (RT-qPCR) and western blotting.” should be included in the end of previous paragraph: 2.1 Culture of Enterococcus faecalis and macrophages because this is an explanation of RAW264.7 culture preparation.

#2. Please provide an explanation how to decide the exposure time (30min prior to infection) and dose (30mol/L) of caspase-1 inhibitor with articles describing those evidence.

#3. Please add the discussion about the reason why the peak PI-positive cell value was higher in lower (1:1, 10:1) MOI infection in the later stage in Figure 1a.

#4. Please include a description of each picture in Figure 1 b-g.

#5. Why the expression of three genes (Caspase-3, GSDMD-N, and MLKL) has decreased in 12h in MOI of 100:1 in Figure 2 b-d?

#6. Why has the number of PI positive cells elevated after 12h even with caspase-1 inhibitor?

#7 Please explain the time point of the picture in Figure3b respectively.

#8. Please add the picture of positive control in Figure 3b.

#9. Please add the explanation why the protein expression of caspase-1 was not affected by adding caspase-1 inhibitor in Figure 5a.

#10. Please make sure the plus and minus sign below the Figure 5a.

#11. In Figure 6, please add the description of experimental group (treatment) for each picture. 

Reviewer 2 Report

Comments are in the PDF. Major revisions are to be done

Round 2

Reviewer 1 Report

The manuscript has been revised well.

Reviewer 2 Report

Revisions are to my satisfaction